# Prepare Linear Distributions with Quantum Arithmetic Units

**DOI:** 10.3390/e26110912

**Published:** 2024-10-28

**Authors:** Junxu Li

**Affiliations:** Department of Physics, College of Science, Northeastern University, Shenyang 110819, China; lijunxu1996@gmail.com

**Keywords:** quantum computing, quantum adders, quantum arithmetic logic units (QALUs)

## Abstract

Quantum arithmetic logic units (QALUs) perform essential arithmetic operations within a quantum framework, serving as the building blocks for more complex computations and algorithms in quantum computing. In this paper, we present an approach to prepare linear probability distributions with quantum full adders. There are three main steps. Firstly, Hadamard gates are applied to the two input terms, preparing them at quantum states corresponding to uniform distribution. Next, the two input terms are summed up by applying quantum full adder, and the output sum is treated as a signed integer under two’s complement representation. By the end, additional phase −1 is introduced to the negative components. Additionally, we can discard either the positive or negative components with the assistance of the Repeat-Until-Success process. Our work demonstrates a viable approach to prepare linear probability distributions with quantum adders. The resulting state can serve as an intermediate step for subsequent quantum operations.

## 1. Introduction

Quantum computing signifies a transformative shift in the landscape of computational capabilities, which introduces a level of computational power that enables the efficient resolution of problems deemed intractable by classical computing paradigms [1,2,3,4,5,6,7,8,9,10,11,12,13]. Quantum arithmetic logic units (QALUs) constitute the centerstone of quantum computing. In classical computing, the elementary arithmetic logic units (ALUs) are designed to execute basic arithmetic operations. Similarly, a variety of QALUs, such as quantum adders, subtractors, and multipliers, serve as the building blocks for more complicated algorithms in quantum computing. The best known example is Shor’s algorithm [14,15], where the quantum modular arithmetic plays a vital role.

Among various QALUs, the quantum adder plays a pivotal role, representing a fundamental operation integral to the execution of all algorithms that are constructed upon it. The first quantum adder was proposed in 1996, when Vedral and coworkers explicitly constructed the first quantum ripple carry adder [16], which is a significant landmark in the development of QALUs. Since then, a variety of quantum adders has been developed, including ripple carry [16,17,18], carry lookahead [19,20], carry save [21], and hybrid [22,23] structures. On the other hand, quantum adders can also be implemented based on quantum Fourier transform (QFT) [24,25]. QFT-based quantum adders [26,27] typically commence with a QFT block that converts the input state to the frequency domain, performs addition using controlled phase gates, and then reverts the state to the original domain through an inverse Fourier transform [28,29]. Moreover, recently, a novel approach was proposed to prepare arbitrary normal distributions in quantum registers, utilizing the QFT-based quantum adders [30].

In this paper, we revisit the fundamental quantum binary adders and propose a practical method for generating a linear probability distribution utilizing quantum adders. The following sections are organized as follows. In Section 2, we revisit the simple implementation of quantum binary adders, and demonstrate the outputs of iterative additions that sum up several standalone uniform superposition states. Next, in Section 3, we present how to prepare the linear probability distribution with quantum full adders in the two’s complement representation. By the end, the discussions and conclusions are present in Section 4.

## 2. Materials and Methods

Like classical binary adders, quantum binary adders add up two given terms, and return the output sum. In Figure 1, we depict the diagram of a typical quantum binary adder. The box colored in blue represents a quantum binary adder, and the inputs are given on the left side, whereas the outputs on the right side. There are 2N qubits initialized at state $|a_{N-1}\cdots a_{0}\rangle$ and $|b_{N-1}\cdots b_{0}\rangle$, representing the input binary numbers ${\left(a_{N-1}\cdots a_{0}\right)}_{bin}$ and ${\left(b_{N-1}\cdots b_{0}\right)}_{bin}$. One qubit is initialized at state |0〉, representing the input carry (in general, the input carry is set as 0). There are *N* qubits all initialized at state |0〉 to store the output sum. Generally, the quantum binary adders do not change the state of the inputs, and the output sum, a N+1 digit binary number, is stored in the other N+1 qubits (including the qubit that represents the input carry). As depicted in Figure 1a, the N+1 qubits, which are prepared at |0〉 at the beginning, are converted to state $|s_{N}\cdots s_{0}\rangle$ after applying the quantum binary adder. The output state represents the output sum ${\left(s_{N}\cdots s_{0}\right)}_{bin}$, and we have
(1)$${\left(s_{N}\cdots s_{0}\right)}_{bin}={\left(a_{N-1}\cdots a_{0}\right)}_{bin}+{\left(b_{N-1}\cdots b_{0}\right)}_{bin}$$
For clarity, here, subscript bin indicates that the number is binary.

In classical electronics, a cascade of one-bit full adders constitutes the simplest adder supporting multiple bits. The one-bit full adder adds three inputs [31]: two binary digits denoted as *a* and *b*, along with an input carry denoted as $c_{in}$. Then, the one-bit full adder produces two outputs: the sum *s* and output carry $c_{out}$. The one-bit full adder performs binary addition as
(2)$$s=a⊕b⊕c_{in}$$
(3)$$c_{out}=\left(a\cdot b\right)+\left(c_{in}\cdot \left(a⊕b\right)\right)$$
where $a,b,c_{in},c_{out},s=0,1$, and $a+b+c_{in}=s+2c_{out}$. Similarly, the quantum one-bit full adder, denoted as $U_{+}$, performs binary addition as
(4)$$U_{+}|a,b,c_{in},0\rangle =|a,b,s,c_{out}\rangle$$
where the outputs *s*, $c_{out}$ are as given in Equations (Equation 2) and (Equation 3). In Figure 2b, we present the circuit of a typical quantum one-bit full adder, which is formed by two Toffoli gates along with three CNOT gates. There are four qubits involved in operation $U_{+}$, two qubits represent the input terms *a* and *b*, one qubit represents the input carry $c_{in}$ and stores the output sum *s*, and the last qubit initialized at state |0〉 stores the output carry $c_{out}$.

The simplest quantum adder that supports multiple bits can be formed by cascading multiple one-bit full adders. Figure 2a is a schematic diagram of three cascade quantum one-bit full adders, where the blue boxes indicate operation $U_{+}$, with inputs on the left side and outputs on the right side. The addition starts from the least significant bits (LSBs) $a_{0}$ and $b_{0}$, and the first operation $U_{+}$ is applied on the corresponding qubits. The output sum $s_{0}$ of the LSB addition is also the LSB of the output sum, whereas the output carry is sent to the addition of the next digit, $a_{1}$ and $b_{1}$. In brief, the output carry performs as the input carry in the addition of the next digit, and the output carry of the most significant bit (MSB) is just the MSB of the output sum.

In general, we can sum up *M* standalone input terms by applying M−1 quantum adders that support multiple bits. Figure 1a is a schematic diagram of summing up three input terms with two quantum adders. At the beginning, all qubits are initialized at state |0〉, and the inputs are prepared at uniform superposition by applying Hadamard gates. Qubits denoted as $q^{A}$, $q^{B}$, $q^{C}$, and $q^{S}$ are involved in the first addition. The output sum is then sent to the succeeding addition, along with qubits $Q^{B}$, $Q^{C}$, and $Q^{S}$. The superscripts indicate that the qubits represent the input terms (A,B), the input carry (*C*), or the output sum (*S*). After adding up *M* standalone input terms, the output quantum state, denoted as $\Phi_{M}$, can be given as
(5)$$\Phi_{M}=\frac{1}{\sqrt{2^{NM}}}\overset{M}{\underset{m=1}{⨂}}\left(\sum_{a_{m}=0}^{2^{N}-1}|a_{m}\rangle \right)⊗|\sum_{m=1}^{M}a_{m}\rangle$$
The *M* input terms are denoted as $a_{m}$, m=1,2,⋯,M. As all inputs are *N* bit binary unsigned integers, $a_{m}$ ranges from 0 to $2^{N}-1$. Then, we measure the qubits that represent the output sum, and denote the probability of obtaining result *x* as Pr(x), where for simplicity, we denote the sum as an integer *x*, and the standard quantum state of the output sum corresponds to the binary form of *x*. The theoretical prediction of Pr(x) for various *M* and *N* are as depicted in Figure 2c–n, where there are in total *M* input terms, and each input is a *N* bit binary number. As *N* and *M* increase, the distribution Pr(x) begins to resemble a normal distribution. Thus, it is a feasible approach to prepare normal distribution with quantum adders! [30] (In Ref. [30], Rattew and coworkers proposed a quantum algorithm for the efficient preparation of arbitrary normal distributions, utilizing the quantum random work, and quantum Fourier Transformations or quantum adders, along with Mid-Circuit Measurement and Reuse (MCMR) techniques. The quantum algorithm can be implemented with a polynomial-depth circuit. Later, in Section 3, we show that our approach can prepare normal distributions also with polynomial-depth quantum circuits.).

## 3. Results

Here, we concentrate on the case M=2, where there are only two input terms. As both the inputs are *N* digit unsigned integers, the output sum is a N+1 digit binary number. Recalling Equation (Equation 5), the overall quantum state after the addition (the input carry $c_{in}=0$) can be written as
(6)$$\Phi_{M=2}=\frac{1}{2^{N}}\sum_{a=0}^{2^{N}-1}\sum_{b=0}^{2^{N}-1}\left(|a\rangle ⊗|b\rangle ⊗|a+b\rangle \right)$$
where the two input terms are denoted as *a* and *b*, and we have $a,b\leq 2^{N}-1$, whereas the output sum, for simplicity denoted as *x*, ranges from 0 to $2^{N+1}-2$. Then, we measure the qubits corresponding to the sum after the addition, and denote Pr(x) as the probability of obtaining result *x*. For $0\leq x\leq 2^{N}-1$, there are in total x+1 pairs of inputs a,b that lead to output sum *x*: a=0,b=x; a=1,b=x−1; ⋯ and a=x,b=0. Similarly, for $2^{N}\leq x\leq 2^{N+1}-2$, there are $2^{N+1}-x-1$ pairs of inputs a,b that lead to output sum *x*: $a=2^{N}-1,b=x-2^{N}+1$; $a=2^{N}-2,b=x-2^{N}+2$; ⋯ and $a=x-2^{N}+1,b=2^{N}-1$. Therefore, the probability of obtaining result *x* can be given as
(7)$$Pr\left(x|c_{in}=0\right)=\left\{\begin{array}{cc} \frac{1}{4^{N}}\left(x+1\right), & \;0\leq x\leq 2^{N}-1\\ \frac{1}{4^{N}}\left(2^{N+1}-x-1\right), & \;2^{N}\leq x\leq 2^{N+1}-2 \end{array}\right.$$
Pr(x) reaches to the peak at $x=2^{N}-1$, and $Pr\left(2^{N}-1\right)=\frac{1}{2^{N}}$. The theoretical predictions of Pr(x) with M=2 are as depicted in Figure 2c,g,k, where the distribution Pr(x) is shaped like an isosceles triangle.

Till now, the output sum has been treated as a N+1 digit unsigned integer. Hereafter, we will demonstrate that the distribution Pr(x) can be converted to a linear distribution with two’s complement representation. Two’s complement is the most common method of representing signed integers in classical computers [31,32]. In two’s complement representation, the value χ of a N+1 bit integer ${\left(x_{N},x_{N-1}\cdots x_{0}\right)}_{bin}$ is given as
(8)$$\chi =-2^{N}x_{N}+\sum_{j=0}^{N-1}2^{j}x_{j}$$
where $x_{N}$ is the sign bit indicating the sign, $x_{N}=0$ for non-negative integers, and $x_{N}=1$ for negative integers. The two’s complement representation of a non-negative number is just its ordinary binary representation, with sign bit 0.

In two’s complement representation, the N+1 digit binary number $\left(x_{N}x_{N-1}\cdots x_{0}\right)$ is treated as a signed integer. For $0\leq x\leq 2^{N}-1$, we have $x_{N}=0$, and the readout χ is still *x*. On the other hand, for $2^{N}\leq x\leq 2^{N+1}-2$, we have $x_{N}=1$, and the readout χ is negative. Recalling that $x=\sum_{j=0}^{N}2^{N}x_{j}$, where *x* is the value of the unsigned binary integer ${\left(x_{N}\cdots x_{0}\right)}_{bin}$, we have
(9)$$\chi =x-2^{N+1},\;2^{N}\leq x\leq 2^{N+1}-2$$
Thus, we can rewrite Equation (Equation 7) in two’s complement representation,
(10)$$Pr\left(\chi |c_{in}=0\right)=\frac{1}{4^{N}}\left|\chi +1\right|,\;-2^{N}\leq \chi \leq 2^{N}-1$$

Consider a simple example, the addition of two 3-digit unsigned binary numbers. In Table 1, we present the output sums for all possible inputs. For each input pairs, there are four components in Table 1. The first component (from top) is the output quantum state, where the input carry $c_{in}$ is 0, and the second component is the corresponding readout of the sum (unsigned/signed). The other two components correspond to the case where $c_{in}=1$, and will be discussed later. For example, consider the addition of |100〉 and |101〉. With input carry $c_{in}=0$, the output quantum state is |1001〉. If we treat the 4-digit binary number ${\left(1001\right)}_{bin}$ as an unsigned integer, then the readout is $1\times 2^{3}+0\times 2^{2}+0\times 2^{1}+1\times 2^{0}=9$. However, if we treat ${\left(1001\right)}_{bin}$ as a signed integer in two’s complement representation, then the readout will be $-1\times 2^{3}+0\times 2^{2}+0\times 2^{1}+1\times 2^{0}=-7$.

In Figure 3, a straightforward demonstration of the addition is present. The quantum circuit is as depicted in Figure 3a, where $q^{A}$ and $q^{B}$ represent the input terms, $q^{C}$ represents the input carry, and the output sum is stored in $q^{S}$. Initially, all qubits are prepared at state |0〉. Hadamard gates are then applied on qubits $q^{A}$ and $q^{B}$, preparing them at uniform superpositions as depicted in Figure 3c,d. Here, in $c_{in}=0$, the operations in the dashed boxes are not applied. Then, the standard quantum adder is applied, summing the input terms, and the qubits $q^{S}$ are measured. If we treat the output sum as an unsigned integer, then the distribution Pr(x) is as depicted in Figure 3e, which is shaped as an isosceles triangle, similar to Figure 2c,g,k. On the other side, if we treat the output sum as a signed integer in two’s complement representation, then we will obtain the distribution as shown in Figure 3g, which is a linear distribution according to Equation (Equation 10).

Notice that in Figure 3g, the probability of obtaining result −1 is zero. Sometimes, we prefer to have the zero probability at 0, in other words, setting Pr(χ=0)=0. For this case, we can set the input carry $c_{in}=1$. In this context, the sum is a+b+1, where 1 is the input carry. Equation (Equation 7) can then be rewritten as
(11)$$Pr\left(x|c_{in}=0\right)=\left\{\begin{array}{cc} \frac{1}{4^{N}}x, & \;1\leq x\leq 2^{N}-1\\ \frac{1}{4^{N}}\left(2^{N+1}-x\right), & \;2^{N}\leq x\leq 2^{N+1}-1 \end{array}\right.$$
The corresponding probability distribution is depicted in Figure 2f. If we treat the output as a signed integer in two’s complement representation, we have
(12)$$Pr\left(\chi |c_{in}=0\right)=\frac{1}{4^{N}}\left|\chi \right|,\;-2^{N}\leq \chi \leq 2^{N}-1$$
The output sums for $c_{in}=1$ are also given in Table 1, where the third component is the output sum with input carry 1, and the last one is the corresponding readout (unsigned/signed). In Figure 2h, the distribution of the probability of obtaining various results after measuring the output sums is depicted, where the output sum is treated as signed integers in two’s complement representation. By this mean, the probability of obtaining result 0 is zero as shown in Figure 2h.

Although the probability is always non-negative, we can also introduce an additional phase to the negative components. As shown in Figure 2b, we can apply a Pauli-Z operation on the qubit $q_{N}^{S}$ after applying the quantum adder. The qubit $q_{N}^{S}$ represents the MSB of the output sum and is the sign bit if we treat the sum as a signed integer in two’s complement representation. Recalling Equation (Equation 8), for negative components, the sign bit is 1; otherwise, the sign bit is 0. Thus, the Pauli-Z operation can introduce an additional phase −1 to the negative components.

Furthermore, we can also discard either the positive or negative components with the assistance of the Repeat-Until-Success process [33,34,35]. After applying the quantum adder, we can measure the qubit $q_{N}^{S}$. If we obtain result |1〉, then the distribution probability is ($c_{in}=1$)
(13)$$Pr_{-}\left(\chi |c_{in}=1\right)=-\frac{1}{\left(1+2^{N}\right)2^{N-1}}\chi ,\;-2^{N}\leq \chi \leq 0$$
Otherwise, if we obtain result |0〉, the distribution probability is
(14)$$Pr_{+}\left(\chi |c_{in}=1\right)=\frac{1}{\left(2^{N}-1\right)2^{N-1}}\chi ,\;0\leq \chi \leq 2^{N}-1$$
where the subscripts ± indicate the measurement results. If we want to obtain a linear probability distribution for the negative range as shown in Equation (Equation 13), then we need to repeat the whole process until we obtain result |1〉 when measuring the qubit $q_{N}^{S}$. On the contrary, if we prefer to obtain the probability distribution for the positive range as shown in Equation (Equation 14), then we need to repeat until we obtain result |0〉.

The time complexity of the proposed approach is mainly determined by the quantum adder. Generally, the time complexity of a typical *N* bit binary quantum adder, as depicted in Figure 2b, is of the O(N) order, where there are 2N Toffoli gates along with 3N CNOT gates. However, due to the limited connectivity, it can be challenging to implement large quantities of Toffoli gates on the quantum computers in NISQ era. To address this issue, we propose an alternative design of quantum binary adders in our recent work [36], utilizing only Pauli-X gates, CNOT gates, and $C\sqrt{X}$ (CSX) gates, and all two-qubit gates are operated between nearest neighbor qubits. The time complexity of the quantum adder is still of the O(N) order. Therefore, to prepare the linear distribution as shown in Equation (Equation 12), the time complexity is O(N). Furthermore, the proposed approach can be extended to prepare normal distributions as depicted in Figure 2c–n. Theoretically, the time complexity of summing up *M* independent uniform superposition states is O((M−1)N). In other words, the proposed approach can be implementable with a polynomial-depth circuit.

## 4. Discussion

In this paper, we revisit the elementary quantum binary adders, and propose a viable approach to prepare linear probability distribution with the quantum adders. There are three main steps to obtain the linear probability distribution. Initially, Hadamard gates are applied to the input qubits, transforming them into uniform superposition states. Following this, the input qubits are processed through a quantum full adder, where their sum is computed, and the resulting output is interpreted as a signed integer in two’s complement representation. At a subsequent stage, a phase shift of −1 is introduced to the components representing negative values. Furthermore, the positive or negative components may be selectively discarded using a Repeat-Until-Success protocol. The resulting output can serve as an intermediate state for succeeding quantum operations.

## Figures and Tables

**Figure 1 entropy-26-00912-f001:**
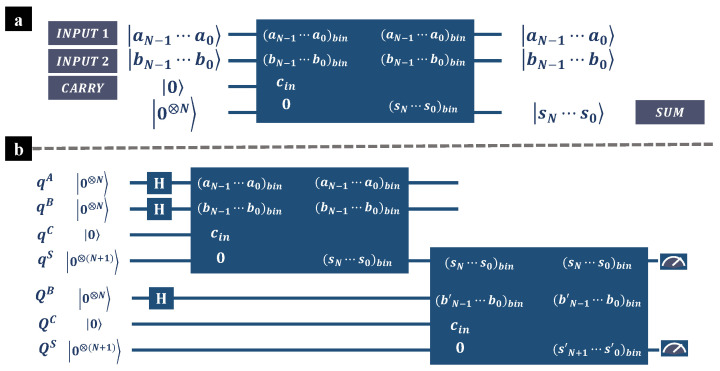
(**a**) Schematic diagram of quantum adder supporting multiple bits. The box colored in blue represents a quantum binary adder, and the inputs are given on the left side, whereas the outputs are on the right side. (**b**) Schematic diagram of summing up three input terms with two quantum adders. At the beginning, all qubits are initialized at state |0〉, and the inputs are prepared at uniform superposition by applying Hadamard gates. Qubits denoted as $q^{A}$, $q^{B}$, $q^{C}$, $q^{S}$ are involved in the first addition. The output sum is then sent to the succeeding addition, along with qubits $Q^{B}$, $Q^{C}$, and $Q^{S}$. The superscripts indicate that the qubits represent the input terms (A,B), the input carry (*C*), or the output sum (*S*).

**Figure 2 entropy-26-00912-f002:**
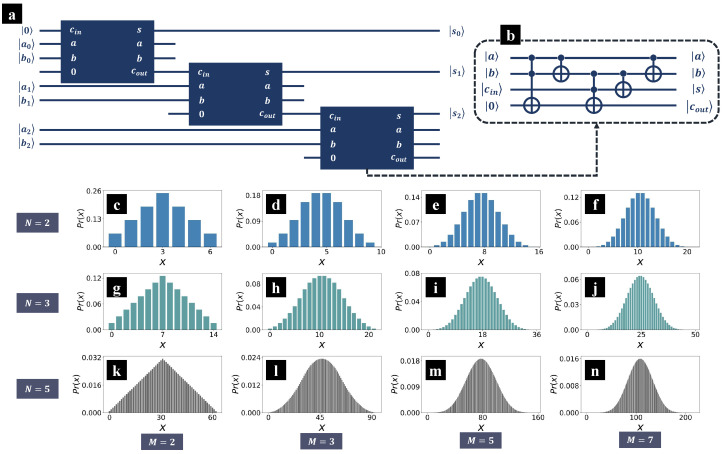
(**a**) Schematic diagram of summing up three input terms with two quantum adders. (**b**) A typical implementation of quantum one-bit full adder. There are in total three CNOT gates and two Toffoli gates. (**c**–**n**) The theoretical prediction of the output sum, where the input terms are *M* standalone uniform superpositions prepared by Hadamard gates. Each input term represents a *N* digit unsigned binary number. Pr(x) is the probability of obtaining result *x*, where we denote the sum as an integer *x*, and the standard quantum state of the output sum corresponds to the binary form of *x*.

**Figure 3 entropy-26-00912-f003:**
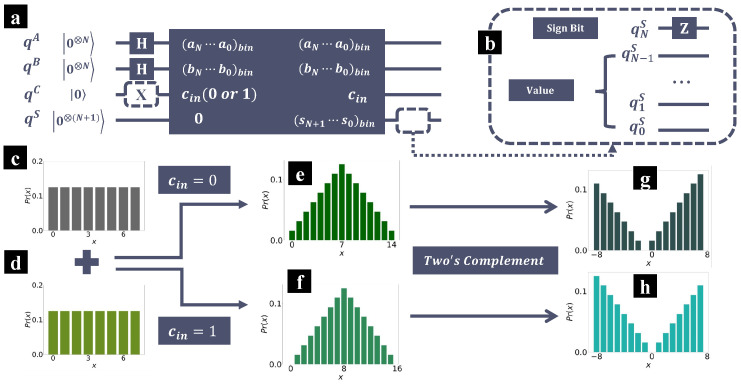
(**a**) Schematic diagram of the quantum circuit that sums up the two uniform superpositions, where $q^{A}$ and $q^{B}$ represent the input terms, $q^{C}$ is the input carry, and the output sum is stored in $q^{S}$. The operations in dashed boxes are applied for $c_{in}=1$. (**b**) To introduce phase −1 to the negative components, we can apply a Pauli-Z gate on $q_{N}^{S}$, which represents the sign bit of the output. (**c**,**d**) Visualization of the uniform superpositions after applying Hadamard gates. (**e**,**f**) Distribution of the probability of obtaining various results after measuring the output sums, which is treated as unsigned integers. (**g**,**h**) Distribution of the probability of obtaining various results after measuring the output sums, which is treated as signed integers in two’s complement representation. In (**e**,**g**) we set the input carry $c_{in}=0$, whereas in (**f**,**h**) $c_{in}=1$.

**Table 1 entropy-26-00912-t001:** Table of the output sums, where both the input terms are 3-bit binary unsigned integers, and the input carry is set either 0 or 1. There are four components in each entry. From top to bottom, the first one is the output sum with input carry 0, and the second one is the corresponding readout (unsigned/signed), whereas the third component is the output sum with input carry 1, and the last one is the corresponding readout (unsigned/signed).

Inputs	|000〉	|001〉	|010〉	|011〉	|100〉	|101〉	|110〉	|111〉
|000〉	|0000〉	|0001〉	|0010〉	|0011〉	|0100〉	|0101〉	|0110〉	|0111〉
	0/0	1/1	2/2	3/3	4/4	5/5	6/6	7/7
	|0001〉	|0010〉	|0011〉	|0100〉	|0101〉	|0110〉	|0111〉	|1000〉
	1/1	2/2	3/3	4/4	5/5	6/6	7/7	8/−8
|001〉	|0001〉	|0010〉	|0011〉	|0100〉	|0101〉	|0110〉	|0111〉	|1000〉
	1/1	2/2	3/3	4/4	5/5	6/6	7/7	8/−8
	|0010〉	|0011〉	|0100〉	|0101〉	|0110〉	|0111〉	|1000〉	|1001〉
	2/2	3/3	4/4	5/5	6/6	7/7	8/−8	9/−7
|010〉	|0010〉	|0011〉	|0100〉	|0101〉	|0110〉	|0111〉	|1000〉	|1001〉
	2/2	3/3	4/4	5/5	6/6	7/7	8/−8	9/−7
	|0011〉	|0100〉	|0101〉	|0110〉	|0111〉	|1000〉	|1001〉	|1010〉
	3/3	4/4	5/5	6/6	7/7	8/−8	9/−7	10/−6
|011〉	|0011〉	|0100〉	|0101〉	|0110〉	|0111〉	|1000〉	|1001〉	|1010〉
	3/3	4/4	5/5	6/6	7/7	8/−8	9/−7	10/−6
	|0100〉	|0101〉	|0110〉	|0111〉	|1000〉	|1001〉	|1010〉	|1011〉
	4/4	5/5	6/6	7/7	8/−8	9/−7	10/−6	11/−5
|100〉	|0100〉	|0101〉	|0110〉	|0111〉	|1000〉	|1001〉	|1010〉	|1011〉
	4/4	5/5	6/6	7/7	8/−8	9/−7	10/−6	11/−5
	|0101〉	|0110〉	|0111〉	|1000〉	|1001〉	|1010〉	|1011〉	|1100〉
	5/5	6/6	7/7	8/−8	9/−7	10/−6	11/−5	12/−4
|101〉	|0101〉	|0110〉	|0111〉	|1000〉	|1001〉	|1010〉	|1011〉	|1100〉
	5/5	6/6	7/7	8/−8	9/−7	10/−6	11/−5	12/−4
	|0110〉	|0111〉	|1000〉	|1001〉	|1010〉	|1011〉	|1100〉	|1101〉
	6/6	7/7	8/−8	9/−7	10/−6	11/−5	12/−4	13/−3
|110〉	|0110〉	|0111〉	|1000〉	|1001〉	|1010〉	|1011〉	|1100〉	|1101〉
	6/6	7/7	8/−8	9/−7	10/−6	11/−5	12/−4	13/−3
	|0111〉	|1000〉	|1001〉	|1010〉	|1011〉	|1100〉	|1101〉	|1110〉
	7/7	8/−8	9/−7	10/−6	11/−5	12/−4	13/−3	14/−2
|111〉	|0111〉	|1000〉	|1001〉	|1010〉	|1011〉	|1100〉	|1101〉	|1110〉
	7/7	8/−8	9/−7	10/−6	11/−5	12/−4	13/−3	14/−2
	|1000〉	|1001〉	|1010〉	|1011〉	|1100〉	|1101〉	|1110〉	|1111〉
	8/−8	9/−7	10/−6	11/−5	12/−4	13/−3	14/−2	15/−1

## Data Availability

All data and materials supporting the results or analyses presented in this paper are available from the corresponding author upon reasonable request.

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
