# Peer review of "Prepare Linear Distributions with Quantum Arithmetic Units"

_entropy, 2024, doi:10.3390/e26110912_

Round 1

Reviewer 1 Report

Comments and Suggestions for Authors

The manuscript proposes a novel quantum algorithm which is able to generate linear probability distributions of numbers represented in binary notation of measured qubits. The algorithm utilises units of quantum adders. The problem of generating specific types of distributions as outputs of quantum computing units is an interesting direction of research, which can find useful applications. There exists a paper (ref. [30]) which proposes a method of generation of Gaussian distributions. The techniques of ref. [30] and the present paper are different, which suggests that the technique of generating arbitrary distribution by feasible means or fixed Quantum Arithmetic Logic Units (QALU) is unknown. The authors of the present manuscript also notice that the system of adders can generate Gaussian distrbutions. In this context it would be interesting to compare complexity of the two techniques. Also, a comment on feasibility of experimental test would be desirable. Therefore, in my opinion the manuscript deserved to be published if the authors make the abovementioned points clearer.

Author Response

We sincerely thank the referee for their critical appraisal of the manuscript and their appreciation for the work! We have carefully revised the manuscript, and all issues are addressed in the attached file.

Reviewer 2 Report

Comments and Suggestions for Authors

This short paper by Li describes a simple but neat protocol to use an aspect of quantum arithmetic to produce linear proability distributions.  It is based on the observation that when adding, say, the two digits on a pair of regular 6-sided dice when thrown at random, the probability of getting numbers 2,3,4,5,6,7 is proportional to the linear sequence 1,2,3,4,5,6 (those being the number of ways of making each sum from the two dice).

Generalising this result, with an indeterminate number of adders, and intepretatation of the in binary representation of the numbers to optionally include a sign bit, the ordering and range of the linear distribution can be manipulated and by selecting on particular outcomes of the sign bit, particular probability distributions can be achieved.

The whole idea is simple, but neat, and the paper explains it clearly.  I recommend publication without change.

Author Response

We sincerely thank the referee for their appreciation for the work!
